

# Numerically consistent budgets of energy, momentum and mass in Cartesian coordinates: Application to the WRF model

Matthias Göbel[1], Stefano Serafin[2], and Mathias W. Rotach[1]

[1]Department of Atmospheric and Cryospheric Sciences, University of Innsbruck, Innsbruck, Austria
[2]Department of Meteorology and Geophysics, University of Vienna, Vienna, Austria

**Correspondence:** Matthias Göbel (matthias.goebel@uibk.ac.at)

**Abstract.** Numerically accurate budgeting of the forcing terms in the governing equations of a numerical weather prediction model is hard to achieve. Because individual budget terms are generally two to three orders of magnitude larger than the resulting tendency, exact closure of the budget can only be achieved if the contributing terms are calculated consistently with the model numerics.

We present WRFlux, an open-source software that allows precise budget evaluation for the WRF model, as well as transformation of the budget equations from the terrain-following grid of the model to the Cartesian coordinate system. The theoretical framework of the numerically consistent coordinate transformation is also applicable to other models. We demonstrate the performance and a possible application of WRFlux with an idealized simulation of convective boundary layer growth over a mountain range. We illustrate the effect of inconsistent approximations by comparing the results of WRFlux with budget

calculations using a lower-order advection operator and two alternative formulations of the coordinate transformation. With WRFlux, the sum of all forcing terms for potential temperature, water vapor mixing ratio and momentum agrees with the respective model tendencies to high precision. In contrast, the approximations lead to large residuals: The root mean square error between the sum of the diagnosed forcing terms and the actual tendency is one to three orders of magnitude larger than with WRFlux. Furthermore, WRFlux decomposes the resolved advection into mean advective and resolved turbulence components,

which is useful in the analysis of large-eddy simulation output.

## 1 Introduction

Budget analysis for variables of a numerical weather prediction model is a widely used tool when examining physical processes in the atmospheric sciences. Energy and mass budgeting has been used, for instance, to understand the governing dynamics of thermally-driven circulations in the mountain boundary layer (Rampanelli et al., 2004; Lehner and Whiteman, 2014; Potter

et al., 2018). Other examples, such as Lilly and Jewett (1990), Kiranmayi and Maloney (2011) and Huang et al. (2018) are listed in Chen et al. (2020). In a budget analysis, the relative weight and the spatial or temporal patterns of individual forcing terms are assessed. For potential temperature, for instance, forcing terms include resolved advection, subgrid-scale diffusion and diabatic processes, such as radiative heating and latent heat release.

Large competing forcing terms adding up to a relatively small total tendency make the budget calculation particularly error-





prone (Chen et al., 2020): If approximations inconsistent with the model numerics are made, the sum of all forcing terms can result in an unclosed budget with a large residual, even if the relative error of each forcing term is small. To obtain reliable budget calculations, Chen et al. (2020) developed a momentum and potential temperature budget analysis tool for the widely used WRF model (Skamarock and Klemp, 2008). Their study focuses on the horizontal momentum budget in idealized 2D simulations of slantwise convection and squall lines, and demonstrates that very small residual values can be achieved by re-

trieving all relevant forcing terms during the runtime of the model. Chen et al. (2020) compare their results with other ways of approximating the momentum budget: neglection of grid staggering, use of a lower-order advection operator and use of the advective form instead of the flux-form of the equations. The authors demonstrate that these approximations strongly deteriorate the budget closure.

Even if the equations in the numerical model are cast in flux-form, some authors use the advective form in budget analyses as

they find it easier to interpret (e.g., Umek et al., 2021). Although it is possible to discretize an advective-form equation in a way that it is numerically equivalent to the respective flux form (Xue and Lin, 2001), the advective form may hinder interpretation when internal processes dominate the budget of the control volume, which can be a single grid cell or a larger volume (Lee et al., 2004).

By design, the budget computation method of Chen et al. (2020) cannot discriminate between tendencies caused by resolved-

scale and subgrid-scale turbulence. This is important, especially when doing large-eddy simulations, which partially resolve the turbulence spectrum. A budget analysis tool capable of estimating tendencies from resolved turbulence must rely on online computation of turbulence statistics during model integration.

Furthermore, budget analysis is more intuitively carried out in the Cartesian coordinate system, while numerical weather prediction models generally adopt a curvilinear terrain-following system. For budget diagnostics in simulation domains with

non-uniform orography, accurate computation of the coordinate transformation between the terrain-following and the Cartesian system is therefore mandatory. This is mainly an issue when tendencies resulting from flux derivatives in a particular spatial direction, such as the vertical derivative of the resolved turbulent flux, are inspected.

Some numerical weather prediction models, e.g. WRF, adopt a mass-based vertical coordinate. Because the atmospheric mass in a model column generally varies during integration, the height of the vertical levels changes with time. Thus, time derivatives

on constant model levels and at constant height are not equal. This also needs to be accounted for if one wishes to compute the total model tendency in the Cartesian coordinate system accurately. When looking at the instantaneous tendencies between individual model time steps, this effect can usually be neglected. However, if the budget is averaged over a time interval, the distance between the vertical levels can change considerably.

The decomposition into mean and turbulent components and the coordinate transformation to the Cartesian coordinate system

were implemented, e.g., by Schmidli (2013) and Umek et al. (2021). However, neither of the two studies aim at a closed budget.

In this study, we present WRFlux, an open-source budget calculation tool for WRF that yields a closed budget, a consistent transformation to the Cartesian coordinate system and decomposition into mean and turbulent components. WRFlux allows to output time-averaged resolved and subgrid-scale fluxes and other tendency components for potential temperature, water vapor





mixing ratio and momentum for the *Advanced Research WRF* (ARW) dynamical core.

The paper is organized as follows. First, we summarize the theoretical foundation of the approach in Sect. 2. This is relevant not only for the WRF model but for any hydrodynamic model in flux-form that utilizes a generalized vertical coordinate. In Sect. 3, details about the implementation of WRFlux are given, followed by the results of an example simulation in Sect. 4.

The purpose of the example simulation is to illustrate a possible application of WRFlux, show its performance and compare it to other, more simplified budget computation approaches.

## 2 Theory

### 2.1 Conservation equation transformations

The flux-form conservation equation for a variable $\psi$ in the Cartesian coordinate system $\boldsymbol{x} = (x_0, x_1, x_2, x_3) = (t, x, y, z)$ reads:


$$\partial_t (\rho\psi) = \sum_{i=1}^{3} -\partial_{x_i} (\rho u_i \psi) + S. \tag{1}$$

where $\rho$ is the air density, $u_i$ are the components of the wind speed vector and $\psi$ is a prognostic variable (e.g., $u_i$, potential temperature $\theta$ or mixing ratio, e.g. of water vapor). The first term on the right-hand side is the advective tendency, the second term contains all other forcing terms for $\psi$.

Eq. 1 can be transformed from the Cartesian coordinate system to general curvilinear coordinates $\boldsymbol{\xi} = (\tau, \xi_1, \xi_2, \xi_3)$. Details about coordinate transformations, especially concerning coordinate systems with a generalized vertical coordinate, can be found for instance in Kasahara (1974), Pielke (1984), Byun (1999) and Liseikin (2010). We only give a short summary here. The transformation of Eq. 1 yields:

$$\partial_\tau (\rho |J| \psi) = \sum_{i=1}^{3} -\partial_{\xi_i} (\rho |J| \nu^i \psi) + |J| S. \tag{2}$$

where $|J|$ is the determinant of the 4x4 Jacobi matrix of the transformation, $J_{ij} = \partial_{\xi_j} x_i$ and $\nu^i = J_{ij}^{-1} u_j = u_j \partial_{x_j} \xi_i$ is the contravariant velocity in the new coordinate system with $\boldsymbol{u} = (1, u, v, w)$. Following Liseikin (2010), we use a four-dimensional coordinate system since the coordinate transformation can be time-dependent.

Many atmospheric models use a coordinate system of the form $\boldsymbol{\xi} = (t, x, y, \eta)$ with the generalized vertical coordinate $\eta$. $\eta$ can be a function of space and time and must possess a monotonic relationship to height $z$ (Kasahara, 1974). The Jacobian matrix

for the transformation to such a coordinate system and its inverse are given in appendix A. The Jacobian determinant reads:

$$|J| = \partial_\eta z \equiv z_\eta. \tag{3}$$

and the contravariant velocity vector is:

$$\boldsymbol{\nu} = (1, u, v, \dot{\eta}) \tag{4}$$





with

$$\dot{\eta} = \partial_t \eta + u\partial_x \eta + v\partial_y \eta + w\partial_z \eta =: \omega. \tag{5}$$

Examples of generalized vertical coordinates include terrain-following coordinates and pressure-based coordinates. WRF, for instance, uses a hybrid terrain-following vertical coordinate based on hydrostatic pressure (Klemp, 2011). In WRF, $\eta$ is a function of space ($x$, $y$ and $z$) and time. The coordinate metric $|J|$ appears as part of the dry air mass $\mu_d = -\rho_d g z_\eta$ in the model equations, where $\rho_d$ is the dry-air density and $g$ the acceleration due to gravity .


Inserting Eq. 3 and 4 into Eq. 2 yields:

$$\partial_\tau \left( \rho z_\eta \psi \right) = \sum_{i=1}^{2} \left[ -\partial_{\xi_i} \left( \rho z_\eta u_i \psi \right) \right] - \partial_\eta \left( \rho z_\eta \omega \psi \right) + z_\eta S. \tag{6}$$

This form of the conservation equations is typically used in numerical weather prediction models. The horizontal and temporal derivatives are taken on constant $\eta$-levels. To make this clear, we continue using $\tau$, $\xi_1$ and $\xi_2$ in the equations, even though

$(\tau, \xi_1, \xi_2) = (t, x, y)$.

For a budget calculation tool, taking the derivatives on constant $\eta$-levels is convenient since it avoids interpolation of the model output to constant height levels. However, we would like to have the individual tendency terms as in the Cartesian coordinate system (Eq. 1) for improved interpretability and for comparison with measurements. To attain both of these requirements we can transform the derivatives in Eq. 1 to be on constant $\eta$-levels. Derivatives with respect to $x_i$ and $\xi_i$ in a coordinate system

with a generalized vertical coordinate are related (Kasahara, 1974; Byun, 1999) as:

$$\partial_{\xi_i} A = \frac{\partial A}{\partial x_j} \frac{\partial x_j}{\partial \xi_i} = \partial_{x_i} A + z_{x_i} \partial_z A \tag{7}$$

with $i = 0, 1, 2$, $j = 0, 1, 2, 3$ and $z_{x_i} := \partial_{\xi_i} z$.

Using Eq. 7 in Eq. 1 yields:

$$\partial_\tau \left( \rho \psi \right) - z_t \partial_z \left( \rho \psi \right) =$$

$$= \sum_{i=1}^{2} \left[ -\partial_{\xi_i} \left( \rho u_i \psi \right) + z_{x_i} \partial_z \left( \rho u_i \psi \right) \right] - \partial_z \left( \rho w \psi \right) + S. \tag{8}$$

The second term on the left-hand side and the second term in square brackets are the correction terms that account for the derivatives being natively computed on constant $\eta$ instead of on constant $z$-levels.

One can show that equation 8 is numerically not consistent with Eq. 6. Numerically consistent means that the budget closes not only analytically but also after discretization. Therefore, we search for an alternative formulation which is equivalent to Eq. 8

and numerically consistent with Eq. 6. Starting with Eq. 6, we first replace $\omega$ with $w$ using:

$$w = \frac{\mathrm{d}z}{\mathrm{d}t} = J_{3j} \nu^j = z_t + z_x u + z_y v + z_\eta \omega. \tag{9}$$





This equation is analogous to the geopotential tendency equation in WRF.

Solving for $z_\eta \omega$, inserting in Eq. 6 and rearranging leads to:

$$\partial_\tau \left( \rho z_\eta \psi \right) - \partial_\eta \left( \rho z_t \psi \right) =$$

$$= \sum_{i=1}^{2} \left[ -\partial_{\xi_i} \left( \rho z_\eta u_i \psi \right) + \partial_\eta \left( \rho z_{x_i} u_i \psi \right) \right] - \partial_\eta \left( \rho w \psi \right) + z_\eta S. \tag{10}$$

Dividing by $z_\eta$ finally yields:

$$z_\eta^{-1} \partial_\tau \left( \rho z_\eta \psi \right) - \partial_z \left( \rho z_t \psi \right) =$$

$$= \sum_{i=1}^{2} \left[ -z_\eta^{-1} \partial_{\xi_i} \left( \rho z_\eta u_i \psi \right) + \partial_z \left( \rho z_{x_i} u_i \psi \right) \right] - \partial_z \left( \rho w \psi \right) + S. \tag{11}$$

Using the product rule and the commutativity of partial derivatives, one can show that equation 11 is equivalent to Eq. 8. However, the correction terms in Eq. 11 (second term on the left-hand side and second term in square brackets) are conceptually different from the correction terms in Eq. 8. While the latter only correct for the derivatives being taken on constant $\eta$-levels, the former also correct for $z_\eta$ being used in the temporal and horizontal derivatives.

In contrast to Eq. 8, Eq. 11 can be closed numerically. For this, we need to close the model equation in the terrain-following coordinate system (Eq. 6) and the geopotential equation (Eq. 9) and $z_\eta \omega$ needs to be numerically equivalent in both equations. The latter two requirements can be achieved by recalculating $w$ based on Eq. 9 instead of using the prognostic value of the model.

## 2.2 The $\theta$-budget

For numerical reasons, WRF uses potential temperature perturbation as a prognostic variable. The perturbation is computed with respect to a constant base state, as $\theta_{\mathrm{p}} = \theta - \theta_0$ with $\theta_0 = 300\,\mathrm{K}$. Based on this decomposition, the energy equation can be split up into advection of the perturbation and of the constant base state:

$$z_\eta S = \partial_t (\rho z_\eta \theta) - \nabla \cdot (\rho z_\eta \boldsymbol{\nu} \theta) =$$

$$= \partial_t (\rho z_\eta \theta_{\mathrm{p}}) - \nabla \cdot (\rho z_\eta \boldsymbol{\nu} \theta_{\mathrm{p}}) + \theta_0 \left[ \partial_t (\rho z_\eta) - \nabla \cdot (\rho z_\eta \boldsymbol{\nu}) \right] \tag{12}$$

with the contravariant velocity $\boldsymbol{\nu}$. Due to the continuity equation, the last term on the right-hand side of Eq. 12 is identically zero. Equation 12 can be used to compute the components of the full-$\theta$-tendency with high numerical accuracy.





## 2.3 Discretization

WRF uses C grid staggering (Arakawa and Lamb, 1977) to discretize the governing equations due to its favorable conservation properties. For the thermodynamic variables (potential temperature and mixing ratio), we discretize equation 11 as:

$$
z_\eta^{-1} \delta_\tau \left( \rho z_\eta \psi \right) - \delta_z \left( \rho z_t \overline{\psi}^z \right) =
$$
$$
= \sum_{i=1}^{2} \left[ -z_\eta^{-1} \delta_{\xi_i} \left( \rho z_\eta u_i \overline{\psi}^{x_i} \right) + \delta_z \left( \rho z_{x_i} \overline{u_i}^{x_i z} \overline{\psi}^z \right) \right]
$$
$$
- \delta_z \left( \rho w \overline{\psi}^z \right) + S \tag{13}
$$

On the right-hand side, the operator $\delta$ denotes central finite differences of the staggered fluxes while overbars denote spatial averaging to the correct location. The averaging operation for $\psi$ depends on the type and order of the advection operator. While the even-order advection operators are spatially centered, the odd-order operators consist of an even-order operator and an upwind term (Skamarock et al., 2019; Chen et al., 2020). In addition to the standard advection operators, WRF offers positive-definite, monotonic and Weighted Essentially Non-Oscillatory options.

For Eq. 13 to be numerically consistent with the conservation equation used in the model (Eq. 6), all terms need to use the same advection operator as in the numerical model. The correction terms derive from the vertical advection term and thus must be discretized in the same way as the vertical advection.

Since the momentum components are staggered themselves the discretization differs from Eq. 13, but still straightforward to derive. We do not state them here for brevity.

In Eq. 8, we introduced a form of the conservation equation that follows immediately from the Cartesian conservation equation but is numerically not consistent with the terrain-following formulation (Eq. 6) used by the model. To demonstrate the inconsistency, we compare Eq.13 with two different discretizations of Eq. 8.

In the first one the corrections for the horizontal derivatives are built by taking the horizontal flux and staggering it horizontally and vertically to the grid of the vertical flux:

$$
\delta_\tau \left( \rho \psi \right) - z_t \delta_z \left( \rho \overline{\psi}^z \right) =
$$
$$
= \sum_{i=1}^{2} \left[ -\delta_{\xi_i} \left( \rho u_i \overline{\psi}^{x_i} \right) + \overline{z_{x_i}}^{x_i} \delta_z \left( \overline{\rho u_i \overline{\psi}^{x_i}}^{x_i z} \right) \right]
$$
$$
- \delta_z \left( \rho w \overline{\psi}^z \right) + S. \tag{14}
$$





This is analogous to the implementation of subgrid-scale diffusion in WRF.

The second one adopts a different discretization in the horizontal correction term to avoid double averaging:

$$
\begin{aligned}
\delta_\tau \left( \rho\psi \right) - z_t \delta_z \left( \rho\overline{\psi}^z \right) = & \\
= \sum_{i=1}^{2} & \left[ -\delta_{\xi_i} \left( \rho u_i \overline{\psi}^{x_i} \right) + \overline{z_{x_i}}^{x_i} \delta_z \left( \rho\overline{u_i}^{x_i z}\overline{\psi}^z \right) \right] \\
& - \delta_z \left( \rho w\overline{\psi}^z \right) + S.
\end{aligned}
\tag{15}
$$

The impact of the approximate budget calculations (Eq. 14 and 15) is discussed in Sect. 4.3.

## 2.4 Flux averaging and decomposition

The decomposition of the resolved fluxes into mean advective and resolved turbulent components requires averaging the fluxes over time and/or space (e.g., Schmidli, 2013). The averaging is considered as an approximation of an ensemble average. Means and perturbations are defined by:

$$
\widetilde{\psi} = \frac{\langle \rho\psi \rangle}{\langle \rho \rangle}, \quad \psi'' := \psi - \widetilde{\psi}.
\tag{16}
$$

$\langle \psi \rangle$ denotes the time and/or spatial block average, $\widetilde{\psi}$ is the density-weighted average and $\psi''$ the perturbation thereof. The decomposition of the resolved flux then reads:

$$
\langle \rho u_i \psi \rangle = \langle \rho \rangle \, \widetilde{u_i}\widetilde{\psi} + \langle \rho u_i''\psi'' \rangle \quad \text{for } i = 1, 2, 3.
\tag{17}
$$

with the total flux on the left-hand side and the mean advective and resolved turbulent fluxes on the right-hand side.

We use the density-weighted average, also known as Hesselberg or Favre averaging (Hesselberg, 1926; Favre, 1969), because WRF is a compressible model. Other studies using density-weighted averaging include Kramm et al. (1995), Greatbatch (2001) and Kowalski (2012). The budget closure is insensitive to whether or not density-weighted averaging is applied. In fact, the latter only affects the partitioning between the mean advective and resolved turbulent fluxes, but not the total flux itself. For typical atmospheric applications, also the impact on the mean advective and resolved turbulent components is hardly noticeable.

The correction flux used in the horizontal corrections in Eq. 11 can be decomposed as:

$$
\langle \rho Z_i \psi \rangle = \langle \rho \rangle \, \widetilde{Z_i}\widetilde{\psi} + \langle \rho Z_i''\psi'' \rangle \quad \text{for } i = 1, 2
\tag{18}
$$

with $Z_i = z_{x_i} u_i$.

## 3 Implementation

We implemented the theoretical framework of the previous section in a diagnostic package for WRF: WRFlux. The main fea-
tures of WRFlux are:





- Budget components are retrieved for potential temperature, water vapor mixing ratio and momentum, including tendencies from the acoustic time step, subgrid-scale diffusion (from all available subfilter-scale models and planetary boundary layer schemes), physical parameterizations and numerical diffusion and damping.

- The fluxes and all budget components except for advection are averaged in time during model integration. The optional spatial averaging and computation of advective tendencies with decomposition into mean advective and resolved turbulent is done in the post-processing. The resolved turbulent component is calculated using Eq. 17.

- The vertical velocity is recalculated with Eq. 9. The last term in Eq. 9 is formulated to be consistent with the vertical advection of the budget variable in the terrain-following coordinate system. To achieve a close match of this recalculated

velocity and the prognostic one, we removed an unnecessary double-averaging of $\omega$ in the vertical advection of geopotential[1]. Except for this modification, which leads to about 10 % stronger updrafts and downdrafts, WRFlux does not change the dynamics of the WRF model.

- To close the budget for both the perturbation $\theta_{\mathrm{p}} = \theta - \theta_0$ and full $\theta$, the last term in Eq. 12 needs to vanish. Since WRF does not actually solve the continuity equation but instead integrates it vertically, this is not trivial. Therefore, we

calculate the temporal term and horizontal divergence terms of the continuity equation explicitly and take the vertical term as the residual. Using the residual has only a marginal effect on the vertical component.

- Dry $\theta$-tendencies can be output even when the model is configured to use moist $\theta$ (Xiao et al., 2015) as the prognostic variable.

- Map-scale factors are taken care of as described in Skamarock et al. (2019). Thus, WRFlux is also suited for real-case

simulations.

- Before each update of WRFlux, an automated test suite is carried out that checks the output of WRFlux for consistency using idealized test simulations with a large number of different namelist settings. Details about the tests can be found in the documentation of WRFlux. The latest version of WRFlux, version 1.3, is based on WRF-ARW version 4.3. WRFlux is easy to install and new releases of WRF are continuously integrated. The post-processing tool is written in Python.

## 4   Example of application and the effect of approximations

### 4.1   Simulation design

We demonstrate the capabilities of WRFlux with a simulation of the diurnal evolution of the convective boundary layer over mountainous terrain using WRFlux version 1.2.1. The model setup, i.e. the initial conditions, terrain specification, grid spacing, land surface properties and the choice of the subfilter-scale model, follows Schmidli (2013).

---

[1]This modification is available as a namelist option starting from WRF version 4.3. See https://github.com/wrf-model/WRF/pull/1338.



WRF's hybrid terrain-following coordinate is used. There are 140 vertical levels with a vertical grid spacing ranging approximately from 8 m at the surface to 50 m at the model top at a height of 5 km. The horizontal grid spacing is $\Delta x = 50\,\mathrm{m}$. The model time step is $\Delta t = 1\,\mathrm{s}$. The domain size is 20 km in the x- (cross-mountain) and 10 km in the y-direction (along-mountain). The boundary conditions are periodic in both directions. The topography is a periodic two-dimensional cosine valley with a flat valley bottom and flat mountain ridge:

$$
h(x) = \begin{cases} h_{\mathrm{m}} & |x| \leq x_1 \\ h_{\mathrm{m}} \left\{ \frac{1}{2} + \frac{1}{2} \cos \left[ \frac{\pi}{x_2 - x_1} \left( |x| - x_1 \right) \right] \right\} & x_1 < |x| \leq x_2 \\ 0 & |x| > x_2 \end{cases} \tag{19}
$$

with the ridge height $h_{\mathrm{m}} = 1500\,\mathrm{m}$ and $x_1 = 0.5\,\mathrm{km}$, $x_2 = 9.5\,\mathrm{km}$. The ridge-to-ridge distance is thus 20 km, equal to the domain width. The distance in $x$-direction is defined to be 0 at the center of the domain.

Implicit Rayleigh damping (Klemp et al., 2008) is used above 4 km. We verified that the damping layer is sufficiently deep to dissipate vertically propagating gravity waves before they could reach the model top, be reflected and cause numerical

instabilities. The advection scheme is 5th-order in the horizontal and 3rd-order in the vertical. Subgrid-scale diffusion follows Deardorff (1980), with different eddy diffusivities for the horizontal and vertical to account for the anisotropic grid.

The boundary layer evolution is driven by a simplified radiation scheme as in Schmidli (2013). The radiative balance at the surface is given by:

$$
R_{\mathrm{n}} = S_{\mathrm{n}} + \epsilon_{\mathrm{a}} \sigma T_{\mathrm{a}}^4 - \epsilon_{\mathrm{g}} \sigma T_{\mathrm{s}}^4 \tag{20}
$$

where $R_{\mathrm{n}}$ is the net radiation, $S_{\mathrm{n}} = 475\,\mathrm{Wm}^{-2}$ is the net shortwave flux, $\epsilon_{\mathrm{a}} = 0.725$ and $\epsilon_{\mathrm{g}} = 0.995$ are the emissivities of the atmosphere and the surface, respectively, $\sigma$ is the Stefan-Boltzmann constant, $T_{\mathrm{a}}$ is the air temperature averaged over the lowest two model levels and $T_{\mathrm{s}}$ is the surface temperature. The remaining components of the surface energy balance—the surface heat and moisture fluxes and the ground heat flux—and the resulting surface temperature $T_{\mathrm{s}}$ are calculated with the NOAH land surface model (Tewari et al., 2004). The surface layer is parametrized with the revised MM5 similarity theory scheme (Jiménez

et al., 2012). The soil type is sandy loam and the roughness length is 0.02 m. With these settings, the spatially and temporally averaged sensible heat flux is roughly $150\,\mathrm{Wm}^{-2}$, similar to Schmidli (2013).

The model is initialized at rest with the lapse rate $\Gamma = 3\,\mathrm{K\,km}^{-1}$. Random initial perturbations of potential temperature drawn from a uniform distribution between -0.5 and 0.5 K are added to the lowest five model levels. The setup leads to negligible latent heat fluxes and a very dry atmosphere, therefore moist processes are neglected. Due to the small domain size and zero

background wind, also Coriolis force effects are not taken into account.

We calculate full $\theta$-tendencies and decompose them into resolved turbulence, subgrid-scale turbulence and mean advective. The averaging in Eq. 16-18 is over 30 minutes and in the y-direction. Due to the y-averaging and the periodic boundary conditions, the flux derivatives in the y-direction are almost zero and therefore not shown. The budget calculation is carried out with the WRFlux procedure (Eq. 13), the two alternative formulations (Eq. 14 and Eq. 15) and with 2nd-order instead of the 3rd and

5th-order advection used by the model.





The budget components are divided by mean density to obtain tendencies of the form $\frac{\langle \partial_t \rho \theta \rangle}{\langle \rho \rangle}$ with units Kelvin per second. This shall not be confused with tendencies in advective form $\langle \partial_t \theta \rangle$.

We quantify the budget closure with the root-mean-square error of the sum of all forcing terms $f$ with respect to the actual model tendency $t$ normalized by the standard deviation of $t$:

$$\mathrm{NRMSE} = \sqrt{\frac{\overline{(t-f)^2}}{\overline{\left(t-\bar{t}\right)^2}}}. \tag{21}$$

The averaging is over all gridpoints and 30-min averaging intervals. Following Chen et al. (2020), we also compute the 99th-percentile of the absolute residual scaled by the 99th percentile of the absolute tendency:

$$\mathrm{r}^{99\mathrm{th}} = \frac{p_{99}(|t-f|)}{p_{99}(|t|)} \tag{22}$$

### 4.2 Cross-valley circulation

We start with a short overview of the individual heat budget components in the example simulation for the averaging period between 3.5 and 4 h after initialization. Since no microphysics scheme is activated and the simplified radiation scheme only affects the surface energy balance, the heat budget in the atmosphere only consists of resolved advection and subgrid-scale diffusion.

Figure 1 shows cross-sections of the total turbulence (resolved + subgrid-scale) and mean advective components of the heat budget in the Cartesian coordinate system (Eq. 13). The dynamics are driven by the surface sensible heat flux; vertical turbulent flux convergence in the layer close to the surface (Fig. 1b) induces upslope winds and compensatory return flows aloft and in the valley center (wind arrows in Fig. 1f). Above the slope wind layer, vertical turbulent entrainment leads to cooling (Fig. 1b). Above the ridge, a convective core develops that transports heat from the surface to higher levels. Lateral turbulent entrainment cools the convective core and warms the surrounding air (Fig. 1a). The mean advective tendency shows regions of horizontal $\theta$-flux divergence on the slope and convergence on the ridge, and vice versa for the vertical (Fig. 1d and e). These regions essentially coincide with regions of mass convergence and divergence (not shown). The scale of the horizontal and vertical advective tendencies is three orders of magnitude larger than for the corresponding turbulence tendencies (Fig. 1a and b). However, the respective sums of the horizontal and vertical components are of comparable magnitude (Fig. 1c and f). Close to the surface, the net mean advective tendency shows cooling of the slope wind layer by the mean upslope wind and warming of the convective core due to horizontal mass convergence (Fig. 1f). The former is weaker than the turbulent heating, while the latter is stronger than the turbulent cooling, leading to net warming close to the surface (Fig. 2a). Away from the surface, the mean advective tendency leads to cooling zones that propagate with time from the ridge towards the valley center. After 4 hours this cooling zone spans almost the whole domain in the horizontal direction (Fig. 1f and 2a).

The total tendency in the mass-based terrain-following coordinate system shows somewhat different structures with stronger warming throughout the domain (Fig. 2b). The only difference between the total tendencies in the terrain-following and the Cartesian formulation is the second term on the left-hand side in Eq. 11, which accounts for the height of the vertical levels being time-dependent. As we can see, this term has a considerable impact and is thus needed to close the budget in Eq. 11. In



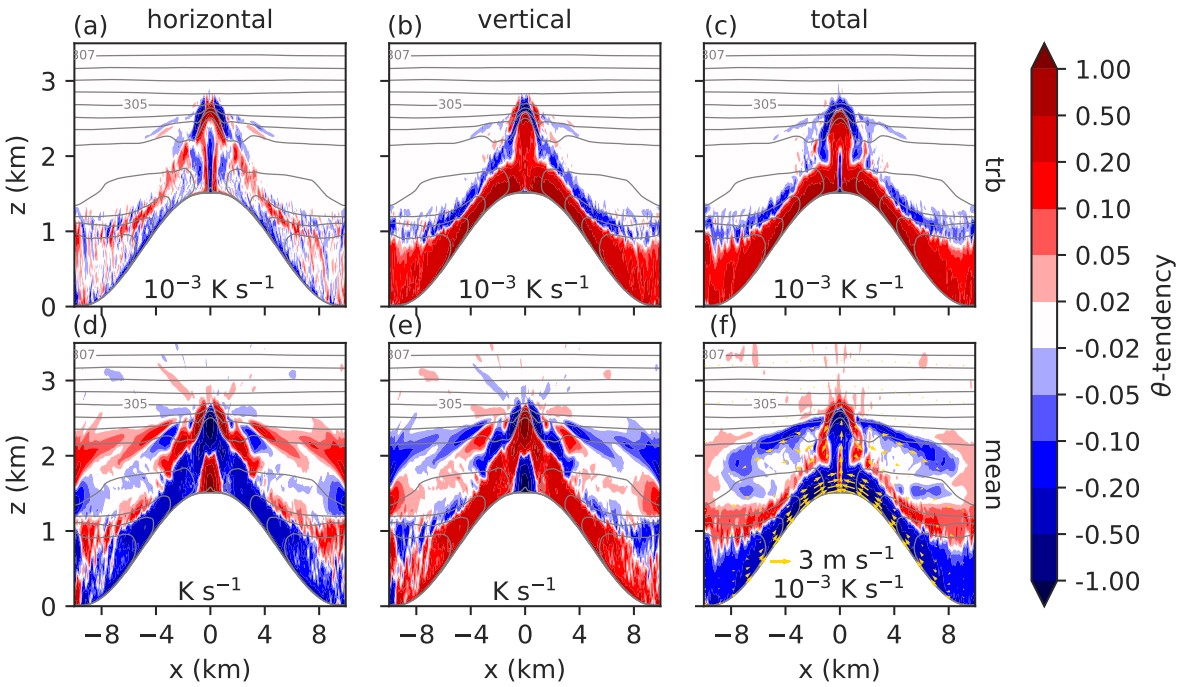

**Figure 1.** Cross-sections of total turbulence (trb = resolved + subgrid-scale turbulence, panels a,b,c) and mean advective (panels d,e,f) $\theta$-tendency components in the Cartesian coordinate system for the averaging period between 3.5 and 4 h after initialization. The calculation is based on Eq. 11, discretized according to Eq. 13 and decomposed with Eq. 17 and 18. The horizontal (panels a and d) and vertical (panels b and e) components are the flux derivatives in the cross-mountain and vertical direction, respectively. The units of the colorbar are denoted in each panel. Note that a different color scale is used in panels (d) and (e). The contour lines represent mean potential temperature with a spacing of 0.5 K. Panel (f) also shows averaged wind vectors.

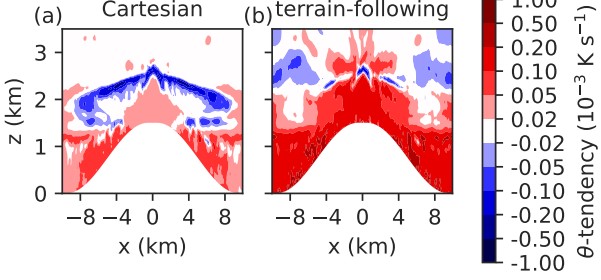

**Figure 2.** Cross-sections of total $\theta$-tendency for the averaging period between 3.5 and 4 h after initialization for the Cartesian (left-hand side of Eq. 11 divided by $\langle \rho \rangle$) and the terrain-following coordinate system (left-hand side of Eq. 6 divided by $\langle z_\eta \rho \rangle$).

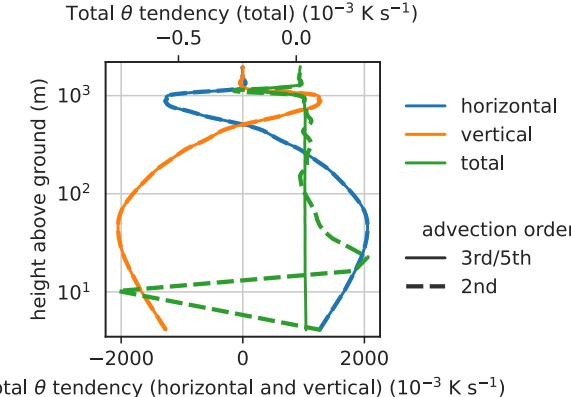

**Figure 3.** Profiles of total $\theta$-tendency (resolved + subgrid-scale) on the ridge at $x = 0$, for the averaging period between 3.5 and 4 h after initialization resulting from flux derivatives in the X (blue, lower x-axis) and Z (orange, lower x-axis) directions and their sum (green, upper x-axis) calculated with 2nd-order advection (dashed) and 3rd/5th-order, consistent with the numerical model (solid).

contrast, in the alternative form of the equation (Eq. 8), the correction term for the time derivative is almost negligible. This shows that the correction terms in Eq. 11 and 8 are conceptually different as mentioned in Sect. 2.1.

Since we use the equations in flux-form, Fig. 1 and 2, in general, cannot be compared to Schmidli (2013), who used the advective form. However, as Schmidli (2013) points out, under the Boussinesq approximation the total turbulence tendency is equivalent in both formulations. In fact, the total turbulence tendency in Fig. 1c is of comparable magnitude and shows very similar spatial patterns as the one in Fig. 6c in Schmidli (2013).

As shown above, a budget equation typically consists of large competing forcing terms that add up to a relatively small total

tendency. To illustrate this, instead of looking at the decomposition into total turbulence and mean advective tendencies as in Fig. 1, we consider the two budget components as they are calculated by the model (not decomposed): resolved advection and subgrid-scale diffusion. The horizontal and vertical components of the resolved advection close to the surface are on the order of $-1 \, \mathrm{K \, s^{-1}}$ and $+1 \, \mathrm{K \, s^{-1}}$, respectively. Their sum is much smaller, on the order of $-10^{-2} \, \mathrm{K \, s^{-1}}$. The subgrid-scale diffusion is on the order of $+10^{-2} \, \mathrm{K \, s^{-1}}$. Adding the resolved advection and the subgrid-scale diffusion leads to a total tendency on the

order of $+10^{-5} \, \mathrm{K \, s^{-1}}$. When adding the large forcing terms, approximations in the budget calculation can lead to considerable errors, as we will demonstrate in the following.

### 4.3 Comparison of budget calculation methods

We compare the budget obtained with WRFlux (Eq. 13) with several alternative forms. The first alternative uses 2nd-order advection in Eq. 13 instead of the advection order that is consistent with the model (3rd/5th-order). The differences between

using 2nd order and the consistent advection order are largest on the ridge, where the vertical velocities are largest. The horizontal and vertical components of the total $\theta$-tendency (resolved + subgrid-scale) in Fig. 3 are both very close for the two

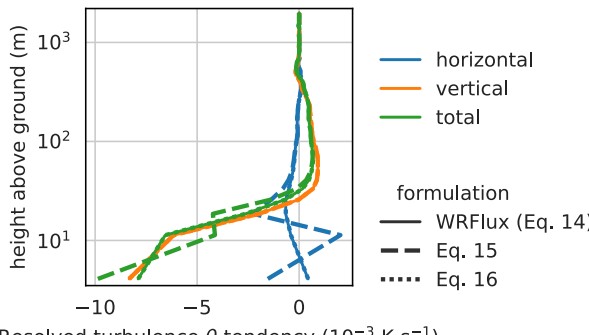

**Figure 4.** Profiles of resolved turbulence $\theta$-tendency over the slope at $x = -5 \, \text{km}$, for the averaging period between 3.5 and 4 h after initialization resulting from flux derivatives in the X (blue) and Z (orange) directions and their sum (green). The line styles indicate different formulations for the horizontal flux derivatives (explained in Sect. 2).

**Table 1.** NRMSE (Eq. 21) and $r^{99}$ (%, Eq. 22) values for all budget variables and budget calculation methods.

|  | $\theta$ | | $q_\mathrm{v}$ | | $u$ | | $v$ | | $w$ | |
|---|---|---|---|---|---|---|---|---|---|---|
|  | NRMSE | $r^{99}$ | NRMSE | $r^{99}$ | NRMSE | $r^{99}$ | NRMSE | $r^{99}$ | NRMSE | $r^{99}$ |
| WRFlux (Eq. 14) | $9.17 \cdot 10^{-3}$ | 1.19 | $1.52 \cdot 10^{-3}$ | 0.18 | $6.44 \cdot 10^{-4}$ | 0.07 | $6.34 \cdot 10^{-4}$ | 0.07 | $5.63 \cdot 10^{-4}$ | 0.06 |
| 2nd-order advection | $1.90 \cdot 10^{0}$ | 234.75 | $3.12 \cdot 10^{-1}$ | 35.20 | $6.17 \cdot 10^{-1}$ | 68.37 | $6.52 \cdot 10^{-1}$ | 69.21 | $8.38 \cdot 10^{-1}$ | 84.96 |
| Eq. 15 | $4.88 \cdot 10^{0}$ | 486.50 | $2.15 \cdot 10^{-1}$ | 22.03 | $3.73 \cdot 10^{-1}$ | 32.11 | $2.12 \cdot 10^{-1}$ | 22.47 | $3.00 \cdot 10^{0}$ | 13.56 |
| Eq. 16 | $1.34 \cdot 10^{-1}$ | 13.36 | $1.63 \cdot 10^{-1}$ | 16.41 | $1.92 \cdot 10^{-1}$ | 22.72 | $1.02 \cdot 10^{-2}$ | 1.10 | $1.18 \cdot 10^{-1}$ | 5.09 |

calculation methods. But when adding these large and opposing components, the 2nd-order calculation yields considerably different results close to the surface. Instead of the constant warming up to the entrainment layer, we see large oscillations of warming and cooling. The main errors derive from the vertical component (not shown). Only the deviation at 250 m above ground originates from errors in the horizontal turbulent entrainment. At other locations in the domain, where the up- and downdrafts are weaker, the differences are smaller.

The different formulations for the horizontal flux derivatives (Sect. 2) differ significantly only where the grid elements are strongly tilted. Figure 4 shows profiles of the resolved turbulence $\theta$-tendency over the slope at $x = -5 \, \text{km}$ for the three different formulations. The calculation of the vertical component is identical for all three formulations. The formulation in Eq. 15, which uses the consistently discretized $\theta$ in the horizontal correction, yields very similar profiles as the reference WRFlux procedure (Eq. 13). In contrast, the formulation in Eq. 14, in which the horizontally destaggered and then vertically staggered horizontal flux is used in the corrections, results in considerable errors close to the surface.

To quantify the differences, we plot the sum of all forcing terms for each budget calculation method against the actual model tendency and compute the normalized root-mean-square error (NRMSE, Eq. 21) in Fig. 5. To avoid averaging out large errors, we drop the spatial averaging for this plot and only use temporal averaging. For the WRFlux procedure (Eq. 13), the points






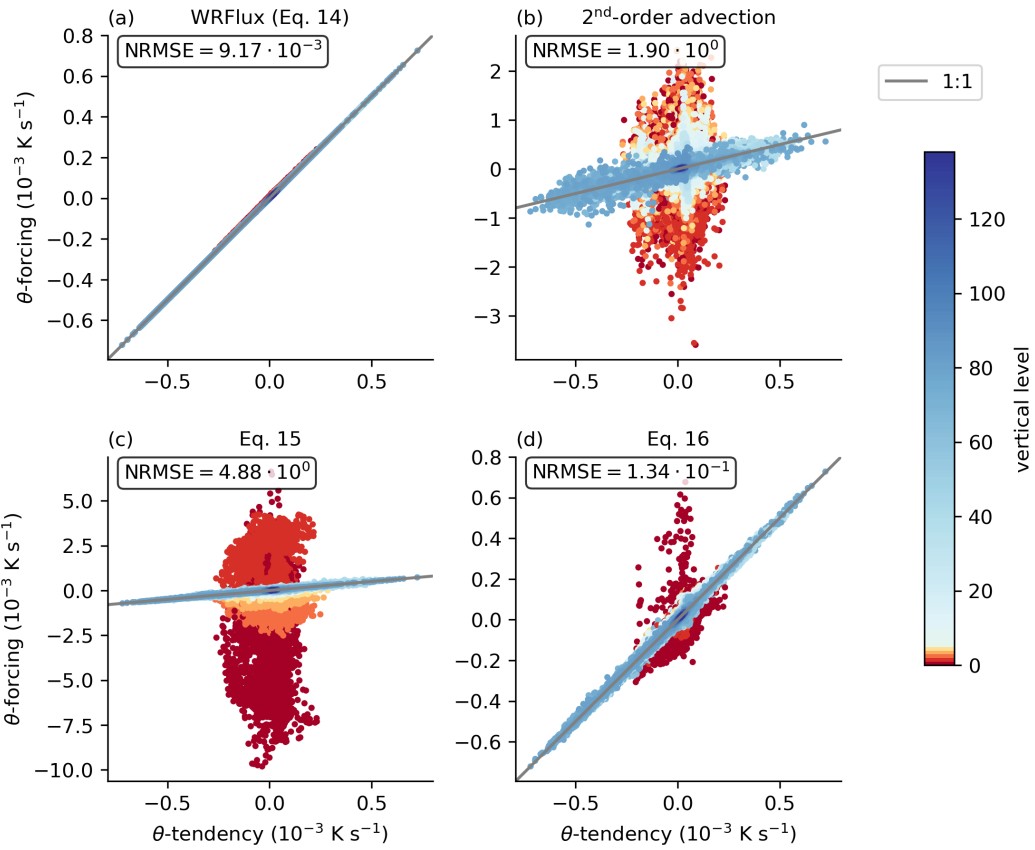

**Figure 5.** Scatter plots of the sum of all $\theta$-forcing terms versus the actual model tendency for all model gridpoints and eight half-hourly (from initialization to 4 h later) values for different budget calculation methods together with the corresponding NRMSE values (Eq. 21). The lower row shows the results for the alternative formulations in Eq. 14 and Eq. 15. For this plot, the data is only averaged temporally, not spatially. The color code indicates the vertical levels of the model gridpoints with a focus on the lowest five levels. The gray line is the 1:1 line that signifies a perfectly closed budget.

lie close to the 1:1 line, indicating a good budget closure, quantified with an NRMSE of $9.17 \cdot 10^{-3}$. The NRMSE increases by about one order of magnitude when using Eq. 15 and by about two orders when using Eq. 14 or 2nd-order advection. The errors are largest at the lowest vertical levels. For water vapor mixing ratio, the NRMSE of WRFlux is about 6 times smaller and for the windspeed components, it is about 15 times smaller (Table 1). For these variables, the other budget calculation

methods lead to NRMSE values that are two to three orders of magnitude worse than the one of WRFlux.

We also tested two other approximations: using WRF's prognostic vertical velocity in the resolved vertical flux instead of the one recalculated with Eq. 9 and not including the density in the time-averaging of the total resolved flux (left-hand side of Eq. 17). The effect on the budget closure is moderate. For both of these approximations, the NRMSE score is increased by about one order of magnitude.





To compare our results to Chen et al. (2020), we compute the 99th-percentile of the absolute residual scaled by the 99th percentile of the absolute tendency (Eq. 22). For potential temperature, WRFlux reaches a value of $r^{99th} \approx 1.2\%$. For horizontal momentum ($u$ windspeed), we reach a score of $r^{99th} \approx 0.07\%$, similar to the value of $0.1\%$ that Chen et al. (2020) state for their simulations.

## 5    Conclusions

We developed a computational method to accurately diagnose the advective and turbulence components of the budgets of prognostic variables in a numerical weather prediction model. The method is based on a numerically consistent implementation of the transformation from a coordinate system with a generalized vertical coordinate, such as a terrain-following coordinate system, to the Cartesian coordinate system. The partitioning of the advective tendency into horizontal and vertical components is different in the two coordinate systems and thus the coordinate transformation is helpful when investigating the horizon-

tal and vertical components separately. We illustrated this by assessing the local heat budget in a simulation of a convective boundary layer over an idealized 2D mountain ridge. The slope flow layer is subject to vertical resolved and subgrid-scale turbulent heating from the ground and turbulent cooling due to vertical entrainment. Close to the surface, the sum of the potential temperature tendencies due to resolved horizontal and vertical advection is about two orders of magnitude smaller than the individual components. Adding the subgrid-scale diffusion yields a final tendency which is another three orders of magnitude

smaller.

   The circumstance of large and counteracting budget components adding up to a relatively small total tendency makes the budget calculation sensitive to approximations. While the sum of all forcing terms in WRFlux agrees to very high precision with the actual model tendency, we could show that approximations based on a lower-order advection operator or a numerically inconsistent formulation of the coordinate transformation lead to large residuals in the budget and noticeable differences in

the tendency profiles. When looking at cross-section plots, the differences between the budget calculation methods are hardly noticeable. Nevertheless, a budget analysis tool that yields large residuals is unreliable. In general, if the residual is large, we do not know whether the individual forcing terms are more or less reliable and only the sum is erroneous or whether also the forcing terms are not trustworthy due to approximations or software bugs. Therefore, a closed budget as achieved by WRFlux is essential. This requires the budget calculations to be consistent with the model numerics.

WRFlux expands the approach of Chen et al. (2020) by the computation of resolved turbulence tendencies and the transformation of fluxes and flux divergence components to the Cartesian coordinate system. Possible applications of our budget analysis tool include the study of

     – the reasons for the unclosed surface energy balance often reported in field studies (e.g., De Roo and Mauder, 2018) for which diagnostics in a layer close to the surface are required;

– the evolution of thermal updrafts in mountainous terrain that are subject to lateral and vertical turbulent entrainment (e.g., Kirshbaum, 2011, 2020);





    – the exchange of heat and moisture between the boundary layer of a valley and the free troposphere (e.g., Rotach et al., 2015; Leukauf et al., 2015, 2017).

Conceivable extensions for WRFlux include a numerically consistent implementation of tendencies in advective form and the
inclusion of further budget variables, such as the mixing ratios of other water species or of a passive tracer.

*Code availability.* WRFlux is available at https://github.com/matzegoebel/WRFlux. The presented example simulation was run with WR-Flux v1.2.1 (Göbel, 2021b) which is based on WRF version 4.2.2. Code specific to the example simulation is deposited at Göbel (2021a). This upload contains the used namelist file, input sounding and modified initialization routine for the LES ideal case in WRF and the simplified radiation scheme introduced in Sect. 4.1.

**Appendix A: Coordinate transformation**

The Jacobian matrix of the transformation from the Cartesian coordinate system $\boldsymbol{x} = (x_0, x_1, x_2, x_3) = (t, x, y, z)$ to a coordinate system $\boldsymbol{\xi} = (\tau, \xi_1, \xi_2, \xi_3) = (t, x, y, \eta)$ with generalized vertical coordinate $\eta$ reads:

$$J = \begin{pmatrix} \partial_\tau t & \partial_{\xi_1} t & \partial_{\xi_2} t & \partial_{\xi_3} t \\ \partial_\tau x & \partial_{\xi_1} x & \partial_{\xi_2} x & \partial_{\xi_3} x \\ \partial_\tau y & \partial_{\xi_1} y & \partial_{\xi_2} y & \partial_{\xi_3} y \\ \partial_\tau z & \partial_{\xi_1} z & \partial_{\xi_2} z & \partial_{\xi_3} z \end{pmatrix} = \tag{A1}$$

$$= \begin{pmatrix} 1 & 0 & 0 & 0 \\ 0 & 1 & 0 & 0 \\ 0 & 0 & 1 & 0 \\ \partial_\tau z & \partial_{\xi_1} z & \partial_{\xi_2} z & \partial_\eta z \end{pmatrix} \tag{A2}$$

which yields the Jacobian determinant $|J| = \partial_\eta z$.

The inverse of $J$ is given by:

$$J^{-1} = \begin{pmatrix} \partial_t \tau & \partial_x \tau & \partial_y \tau & \partial_z \tau \\ \partial_t \xi_1 & \partial_x \xi_1 & \partial_y \xi_1 & \partial_z \xi_1 \\ \partial_t \xi_2 & \partial_x \xi_2 & \partial_y \xi_2 & \partial_z \xi_2 \\ \partial_t \xi_3 & \partial_x \xi_3 & \partial_y \xi_3 & \partial_z \xi_3 \end{pmatrix} = \tag{A3}$$

$$= \begin{pmatrix} 1 & 0 & 0 & 0 \\ 0 & 1 & 0 & 0 \\ 0 & 0 & 1 & 0 \\ \partial_t \eta & \partial_x \eta & \partial_y \eta & \partial_z \eta \end{pmatrix} \tag{A4}$$



*Author contributions.* MG developed the theoretical framework and the software package, ran and analyzed the example simulation and
prepared the manuscript. SS and MWR supervised the work and gave extensive feedback on the manuscript.

*Competing interests.* The authors declare that they have no conflict of interest.

*Acknowledgements.* This work is funded by the Austrian Science Fund (FWF) research project P30808-N32 "Multiscale Interactions in
Convection Initiation in the Alps".

The computational results presented have been achieved using the Vienna Scientific Cluster (VSC).
We would like to thank Lukas Umek, Alexander Gohm and Wiebke Scholz for their suggestions and the many fruitful discussions and Lukas
Umek for providing his code (published in Umek et al., 2021) which was the starting point of WRFlux.





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
