# Peer review of "Numerically consistent budgets of potential temperature, momentum and moisture in Cartesian coordinates: Application to the WRF model"

_Geoscientific Model Development, 2021_

## Author Response (AR1)

**Author response**

As we included a new equation and a new figure, the equation numbers and figure numbers have changed. Line, equation, and figure numbers in our replies refer to the revised manuscript.

**Reviewer 1:**

Thank you very much for your positive review and your comments!

Here are our replies:

**1. Line 235. It would be good to mention the length of the simulation here as this surface flux would only be reasonable for a period of a few hours.**

We added the simulation time (4 h) in the following paragraph.

**2. Figure 2 caption. Should this be Eq. 8 instead of 6?**

Eq. 6 is correct. It refers to the budget equation in the terrain-following coordinate system as it appears in the WRF model equations.

**3. Line 292-295. Some orders of magnitude here seem ten times too large compared to Figures 1 and 2. 10^(-3) and 10^(-4) look more accurate.**

The +/-1e-2 numbers refer to SGS and resolved contributions, respectively. These are not shown separately in any figure. Fig. 2 only shows the decomposition into total turbulent (resolved + SGS) and mean advection.

The total tendency close to the ground above the ridge can be seen in the solid green line in Fig. 4. It is about 0.04e-3=4e-5. This is why we wrote on the order of 1e-5.

**4. 5. and 7.:**

Yes, indeed. However, since we included an additional equation (Eq. 12) in the revised manuscript, the labels are correct now.

**6. Line 303. Is this the lowest level only?**

At the lowest mass level (~3m agl) the tendency in the green dashed line is positive, at the second level (~9m agl) it is strongly negative, and at the third level (~15m agl) it is strongly positive again. But also higher up the differences between the two green lines are rather large, at least up to the 15th vertical level (~100m agl). The AGL height on the y-axes of Fig. 4 and 5 had a small error in the computation. We have corrected these figures.

**8. Line 318. References to Eq. 15 and 14 should be 16 and 15.**

The references are correct here.

**9. Line 339. Maybe "total" is better than "final" here.**

ok

**10. Figure 2. Comment: This large difference is interesting and I would like to have a better conceptual idea of why. Is it because the Cartesian representation is somehow less sensitive to coordinate motion? In Fig. 2b are we looking at the expansion of the coordinate layers with heating?**

What we see in Fig. 3b is the LHS of Eq. 6. This is not the rho*theta tendency on constant model levels, but the tendency of rho*z_eta*theta (governing equation in WRF) normalized with mean(rho*z_eta).

If we would plot the rho*theta tendency on constant model levels (1st term on LHS of Eq. 8) it would look very similar to Fig. 3a. So yes, the difference between Fig 2a and 2b arises mainly from the expansion of the coordinate layers with heating (magnitude of z_eta increases with time). And yes, the representation in Fig. 3b is more sensitive to coordinate motion.

**Reviewer 2**

We are grateful to the reviewer for her/his insightful and constructive comments.

1. **As I understood it, the current implementation of WRFlux needs it to run online with WRF. I wonder if there is a way to provide an offline version without significant changes to the WRF code, which will be much easier to use for most people.**

   Budget analysis on WRF simulations requires two steps: (1) Sending individual tendency terms to the WRF output stream; (2) Recursively averaging all relevant quantities over time to enable an estimation of the explicitly resolved turbulent fluxes. WRFflux implements the necessary code changes for both operations. Doing the same analysis entirely offline is not impossible in principle, but would not be efficient. As an alternative to step (1), one could develop code to compute budget terms from regular model output, mimicking exactly what WRF does; but this would imply rewriting large parts of WRF's dynamical core and parameterizations. As an alternative to step (2), one could output every single time step during the model integration and then do the averaging offline; but this would require immense storage space, besides being computationally inefficient in comparison to recursion. We believe our solution is an optimal compromise between the complexity of the task and the usability of the tool. The GitHub page (https://github.com/matzegoebel/WRFlux/) includes an extensive manual and we are open to help users getting the tool to run.

2. **Section 2.1, Equation 11: I think it would be helpful to readers if a reference is made to the fact that all the WRF prognostic variables are so-called "coupled" (multiplied by the mass inside the grid cube per unit area) as explained in the technical notes?**

   We added this in line 96.

3. **Line 195: "The fluxes and all budget components except for advection are averaged in time during model integration". I don't know what "fluxes" refer to here. Surface fluxes?**

   "Fluxes" refers to the subgrid-scale and resolved fluxes of the five prognostic model variables in the whole domain, not only at the surface. We changed this in line 212. The "except for advection" only refers to the budget components.

4. **Line 280: "The only difference between the total tendencies in the terrain-following and the Cartesian formulation is the second term on the left-hand side in Eq. 11, which accounts for the height of the vertical levels being time-dependent". I wonder if the averaging contributes to the difference as well. For the first term in Eq. 11, the averaging is done after dividing by z_eta.**

   The coupled tendency is first calculated with the time-averaged flux in Eq. 10 and then divided by $\langle \rho\, z_\eta \rangle$. This was not stated clearly in the original manuscript, so we corrected the caption of Fig. 3.

5. **The equation numbers are probably wrong in the figure/table labels, shouldn't they be Eqs. 13, 14 and 15 rather than 14, 15 and 16?**

Yes, indeed - thank you for spotting this. However, since we included an additional equation (Eq. 12) in the revised manuscript, the labels are correct now.

6. **Line 313: "To quantify the differences, we plot the sum of all forcing terms for each budget calculation method against the actual model tendency ..." By that sentence do you mean plotting the LHS and RHS of Eqs. 13, Eqs. 13 with 2nd-order advection, 14 and 15 respectively? If so, It would be clearer to say that explicitly.**

   Yes. We modified lines 347-348 and the caption of Fig. 6 accordingly.

**Reviewer 3**

We are grateful to the reviewer for her/his insightful and constructive comments.

Specific comments

1. **Title: Suggest changing "energy" and "mass" to "potential temperature" and "moisture", respectively. Most results are about the θ tendency budget, which is related but not equivalent to "energy budget" in the strict sense. The current title may misleadingly imply an Earth's energy budget, moist static energy budget, turbulence kinetic energy (TKE) or total energy budget, etc. Furthermore, I don't think the result of the "mass budget" is ever shown.**

   OK, we changed the title accordingly.

2. **Abstract: I suggest reconstructing the abstract to emphasize the unique features and applicability of WRFlux to differentiate this work from Chen et al. (2020), such as the retrieval of the resolved turbulence component, transformation to the easy-to-interpret Cartesian coordinate, etc., all of which are particularly useful for large-eddy simulations or simulations over non-uniform topography.**

   Good point. To emphasize these features, we added a sentence and moved the last sentence to a more prominent place (L5-9).

3. **Introduction: It is important to mention that there have been some attempts to achieve a precise budget retrieval in the exact WRF model (Chen et al. 2020 is not the first), e.g., Lehner (2012, https://collections.lib.utah.edu/ark:/87278/s61n8fxw), Moisseeva and Steyn (2014, https://doi.org/10.5194/acp-14-13471-2014) and Potter et al. (2018, https://doi.org/10.1029/2018JD029427), etc.**

   OK, we added these references in the introduction in L28-29.

4. **L35-38: I'm not sure how the advective form may "hinder interpretation when internal processes dominate the budget of the control volume, which can be a single grid cell or a larger volume"? Can you please elaborate? We added a sentence there: "For instance, if the fluxes on two opposing sides of a grid box are equal, the flux form yields zero net tendency, while this is not necessarily the case for the advective form." (L39-40)**

Lee et al. 2004 looked at the budget of a larger control volume and noted that this is due to "internal processes" in the control volume. We don't want to explain this further in the paper, so we dropped the unclear mentioning of the "internal processes".

5. **Chen et al. (2020) noted that a closed budget for w is challenging, partially because w equation is implicit, coupled with the geopotential tendency equation, and partially due to the variable's inherent rapid variation on small scales. How does WRFlux overcome this issue for the w budget? It is also surprising to see that WRFlux performs the best for w compared to all the other variables and that the accuracy (in terms of NRMS and r99; Table 1) is much higher for the momentum than for the thermodynamic variables ($\theta$ and $q_v$) in your case. Do you have any idea why? Is this result case-dependent or relevant to the physical design of WRFlux? Chen et al. found that closing the w-budget offline is difficult or impossible due to the mentioned issues, but the online budget calculation works fine for them. They also achieved a higher degree of budget closure for w than for horizontal momentum. Our budget calculations are also done online considering all budget components for w, including the important acoustic-step contributions.**

 In contrast to Chen et al. (2020), our budget calculations are time-averaged. To check the effect of the time averaging on the budget closure, we did two small test simulations with and without time averaging. Without time averaging, the NRMSE scores in the terrain-following coordinate system become considerably better for $q_v$ and w and slightly worse for $\theta$, u, and v.

 We did not investigate why the budget closure in the shown simulations is higher for momentum than for $\theta$ and $q_v$. However, in some of our test simulations, it was the other way around, so it is definitely case-dependent (background wind speed, moisture content, physical parameterizations used, etc.).

6. **It would help readers follow Section 2 more easily if the connections between subsections are made more explicit and some terminologies are explained more precisely. - For example, I find it confusing that "Eq. 11 is equivalent to Eq. 8" (L124-128), but "Eq. 8 cannot be closed numerically in contrast to Eq.11" (L128). This confusion may be related to how you define the term "close" here. Suppose you are referring to "budget closure". In that case, my understanding is that it depends on the accuracy of the conducted budget calculation, not the format of the chosen equation to which the budget analysis is applied. Do you mean that because Eq. 11 is numerically consistent with the equation used in the model, the left-hand-side terms of Eq. 11 represent more accurately the actual tendency simulated by the model? And therefore, a budget analysis using Eq. 11 can achieve a higher degree of budget closure? Please clarify.**

 Eq. 11 is similar to WRF's governing equations because the coordinate metric $z_\eta$ appears within the derivatives. In the manuscript, we showed how the budget can be closed with Eq. 11. When using the product rule to transform Eq. 11 to Eq. 8, two terms cancel out analytically, but not numerically. This is because one of the canceling terms comes from a horizontal derivative and the other from a vertical derivative. Therefore, the right-hand side of Eq. 16 is not identical to the right-hand side of Eq. 14. Thus, we argue that Eq. 16 cannot be closed numerically in WRF. The left-hand sides of Eq. 14 and 16, instead, are almost identical in our simulations.

 To clarify this point, we added Eq. 12 and a bit of explanation in L167-170.

 **- Also, I suggest reminding readers at the beginning of Section 2.4 that Eq. 11 (i.e., Eq. 13 after discretization) will be used for the precise budget calculation. It would also be helpful to specify which terms in Eq. 11 can be decomposed into the mean advective and turbulent components (all right-hand-side terms excluding S or all right-hand-side terms excluding only some partial S, i.e., the subgrid physics?)**

OK. We added a few introductory sentences in Section 2.4 with the suggested content (L184-189).

7. **L113-L114: "Numerical consistent" --> "Numerical consistency". Also, placing this sentence ("Numerically… after discretization.") here seems odd. You are neither showing the discretization nor discussing the budget closure in the following lines yet.**

   That's right. Nevertheless, in our opinion, this sentence is important for the logical train of thought. We added a reference to the discretization section in L116.

8. **L130: It is still unclear to me why recalculating the vertical velocity is desired. If Eqs. 6 and 9 are also used in WRF, doesn't it make more sense to use WRF's prognostic to be numerically consistent with the model? Unless this is relevant to the issue I mentioned in comment #4?**

   Recalculating w is necessary to obtain a closed budget in the Cartesian coordinate system. Originally, there was a significant difference in the calculation of $z_\eta \omega$ in WRF's implementation of Eq. 6 and 9. We reduced this difference strongly by changing the discretization of Eq. 9 in WRF (L216-221, https://github.com/wrf-model/WRF/pull/1338). Nevertheless, there is still a small difference remaining: In Eq. 6, $z_\eta$ is included in the coupled prognostic variable $\mu_d \omega$, whereas in Eq. 9 it is calculated directly from the geopotential. To overcome this difference and obtain a closed budget, we recalculate w in a consistent way and use it in the vertical advection term. Whether this is appropriate is definitely debatable. But the effect on w itself is hardly noticeable. Only when adding up the large and counteracting budget components, the difference becomes noticeable. With the prognostic w, the budget closure (NRMSE) is about one order of magnitude worse (L356-359). We suppose the reviewer wanted to ask whether this is relevant to the issue mentioned in comment #5, not #4. These issues are not connected. The recalculation of w mentioned is only relevant for the transformation to the Cartesian coordinate system. It is not used as the prognostic variable in the conservation equation for w. We clarified this in L217-219.

9. **L191-194: Does this part follow exactly Chen et al. (2020)?**

   The retrieval of subgrid-scale tendencies is different since we only extract the fluxes and then do the tendency calculation offline (as for the resolved fluxes). The retrieval of other forcing terms (physics, pressure gradient, damping, diffusion, Coriolis) was implemented very similarly to Chen et al. but independently from them.

10. **L195: "The fluxes and all… are averaged in time during model integration". Averaged over the entire simulation? Or over a user-specified time window? If so, what are your suggested value and the physical reason behind it?**

    Over a user-specified time window (we added this in L213). In this study, we use 30 min (L272) as is often used for Reynolds averaging (compromise between large sample for statistics and stationarity). We added a sentence with a reference on this in L272-274.

11. **L203-204: "Since WRF does not actually solve the continuity equation..." This may be misleading. WRF does solve the continuity equation, just not in the same format as expressed in your Eq. (12), i.e., in terms of the 3D variable $\rho$ but column dry air mass per unit area (2D $\mu_d$). Note that $\mu_d$ is indeed advanced with time in the WRF dynamical solver. I suggest changing the sentence to something like "Since WRF does not directly solve for $\rho$ but the integrated column dry air mass…"**

    Actually, WRF's mass continuity equation in terms of $\mu_d$ (Eq. 2.12 in the technical notes version

4, http://www2.mmm.ucar.edu/wrf/users/docs/technote/v4_technote.pdf) does correspond to the rightmost term in our Eq. 13, except for a factor -g. In fact, $\mu_d$ and $\rho_d$ are related by a diagnostic equation: $\mu_d = -g \rho_d z_\eta$. Ensuring that the last term of our Eq. 13 is exactly zero is difficult because the mass continuity equation in WRF is indeed solved but not integrated explicitly in the acoustic time steps, but rather $\mu_d$ is diagnosed from its definition (Eq. 3.20 in the technical notes) after vertically integrating the mass divergence (Eq. 3.22). We acknowledge that our original formulation was misleading, and we replaced it with the information given in the preceding sentence (L223-225).

12. **L244: "The setup leads to … a very dry atmosphere, therefore moist processes are neglected." Not sure if I overlook something but the initial setup of the moisture field is not mentioned?**

   Yes. We added the specification of the moisture profile (constant RH=40%) in L263.

13. **L246: "We calculate full θ-tendencies and decompose them into resolved turbulence, subgrid-scale turbulence and mean advective." What about the rest of the retrieved budget terms, such as "physical parameterizations and numerical diffusion and damping"? To clarify, it may help to move the sentence "Since no microphysics scheme is activated and the simplified radiation scheme only affects the surface energy balance, the heat budget in the atmosphere only consists of resolved advection and subgrid-scale diffusion" (L261-263) here and mention that for general applications, other grid-resolved parameterized physics terms are possible and categorized as additional budget components.**

   We moved the sentence as suggested and included the suggested additional sentence in L268-270.

14. **Figure 1: Panels a-c show the total turbulence (trb), which is the sum of the resolved and subgrid-scale components. I'm interested in their individual contributions. E.g., What is the relative magnitude between these two components? Do they have similar spatial distributions with the same signs, or do they offset each other? Considering that one of the distinctive features of this tool from previous WRF budget retrieval works is the decomposition into mean and turbulent components, I suggest strengthening the relevant discussion.**

   The subgrid-scale component is mainly relevant very close to the surface and has a similar magnitude but opposite sign as the resolved turbulence component. We added a profile plot showing this decomposition (new figure 2) and some explanation in L294-300.

15. **L283: "...in the alternative form of the equation (Eq. 8), the correction term for the time derivative is almost negligible" I'm lost. This is not shown or can be inferred by figure 2?**

   Yes, this is not shown (we added "not shown" in L316). The plot for Eq. 8 looks the same as Figure 3a. But the correction terms on the LHS of Eq. 8 and 11 are conceptually different (L134-136) since the prognostic variable is different in both equations ($\rho\theta$ vs $\rho z_\eta\theta$).

16. **L290-295: Have you checked if the sum of all the budget components in this alternative analysis is in close balance with the sum of your Eq. 11(13) in Fig. 1? Fig. 1 does not show the total tendency, only the turbulent and mean advective components, not the sum of both. The "total" in the plot refers to the sum of horizontal and vertical components. We added this clarification in the caption.**

   The total tendency is shown in Fig. 3a. Close to the surface on the ridge the tendency is about

0.03 x $10^{-3}$ so on the order of $10^{-5}$ K s$^{-1}$ as written in L329 and visible in Fig. 3a.

**Technical corrections:**

- **L113, L124, …: Replace "equation" with "Eq." here. Please check the rest of the manuscript for consistency.**

  Thanks, we fixed this at several locations.

- **L134: "energy" --> "potential temperature"**

  OK

- **L154-155: "Since… to derive." This sentence is confusing. Suggest changing to "Although the momentum variables are staggered differently from the thermodynamic variables, their discretized equations can be derived analogously…"**

  OK

- **L226: "ridge-to-ridge" --> "bottom-to-bottom" or "valley-to-valley" ?**

  OK, we changed it to "valley-to-valley".

- **Figure 4 legend: I believe the legends "WRFlux (Eq. 14)" should be replaced by "WRFlux (Eq. 13)", "Eq. 15" by "Eq. 14", and "Eq. 16" by "Eq. 15".**

  Yes. However, since we included an additional equation (Eq. 12) in the revised manuscript, the labels are correct now.

- **Figure 5 subtitles for each panel: Same issue as above.**

  Yes

- **Table 1: Same issue as above.**

  Yes

---

## Referee Report (RR1)

The authors have addressed my concerns in the previous review and the modified manuscript with the added texts and figure is now clearer and provide more insights. I recommend the paper to be published after clarifying the minor points listed below.

- Clarify "numerical consistency" in Section 2:
  Following L116 ("Numerical consistency means that budget closes not only analytically but also after discretization"), these two sentences "Eq.8 is numerically not consistent with Eq. 6" (L116) and "we search for an alternative formulation which is… numerically consistent with Eq. 6 " (L117-L118) really don't make sense. First, Eq. 8 and Eq. 6 are two mathematically equivalent, continuous differential equations, and saying that they are not numerical consistent is confusing and meaningless as numerical scheme/analysis has not come into play yet (In contrast, L160-L161 makes more logical sense as you're discussing the numerical methods used to solve the equations there). Furthermore, budget closure generally refers to the balance between the left- and right-hand sides of ONE equation. Therefore, when indicating two equations are numerically consistent or not with your given definition linking to budget closure, it is unclear whose budget closure you are implying. It appears to me that two concepts are somewhat mixed up: one is budget closure, which is theoretically possible for any form of equation as long as the applied numerical analysis of that equation showing a balance between its left- and right-hand side terms. The other concept is "numerical consistency", which I guess you meant which derived equations in the Cartesian coordinate, after discretization, can be more consistent to the discretized governing equation solved in the model (i.e., compared to Eq.8, Eq. 11 is more similar to Eq. 6 because "the coordinate metric $z_\eta$ appears within the derivatives" from the authors' reply to my previous comment #6). The required modifications are minor but It is important to be precise here as these sentences are key to understand why an alternative equation is necessary and to justify your selected budget equation for the precise budget tool. I also found that these confusing wording disturb the logical flow of this article. Below are some possible changes that authors may consider (please modify the content if I misunderstood anything):

L116: "As will be pointed out in Sect. 2.3, Eq. 8 is not ideal for budget closure because the contained derivative terms cannot be discretized using consistent numerical methods with those

for the governing equation (Eq. 6) in WRF."

L128: "Using the …., one can show that Eq. 11 is mathematically equivalent to Eq. 8. For example, the horizontal flux divergence term in Eq. 10 can be expressed as:
(Eq. 12)
Dividing Eq.12 by $z_\eta$ gives the same expression of the horizontal flux divergence term in Eq. 8. The left-hand side of …analogously."

L137: "Instead of Eq.8, we select Eq.11 as the budget equation because the coordinate metric $z_\eta$ appears within the derivatives as in the WRF governing equation (Eq. 6), and so the associated budget analysis can be closes more preciously in consistent with the model dynamics (see Sect. 2.3). For this, we need to …

- L269-270: "For general applications, other grid-resolved and parameterized physics terms are possible and categorized as additional budget components." I acknowledge that this sentence was my suggestion but I missed an "and" there in my previous review comment…

---

## Author Response (AR2)

Author response to reviewer report #2:

We thankfully adopted all the suggestions of the reviewer in almost the exact same wording.